# Enhancing Off-Road Topography Estimation by Fusing LIDAR and Stereo Camera Data with Interpolated Ground Plane

**DOI:** 10.3390/s25020509

**Published:** 2025-01-16

**Authors:** Gustav Sten, Lei Feng, Björn Möller

**Affiliations:** Engineering Design, KTH Royal Institute of Technology, SE-100 44 Stockholm, Sweden; bjornm@md.kth.se

**Keywords:** sensor-fusion, topography estimation, ground interpolation, Kalman filter, off-road navigation

## Abstract

Topography estimation is essential for autonomous off-road navigation. Common methods rely on point cloud data from, e.g., Light Detection and Ranging sensors (LIDARs) and stereo cameras. Stereo cameras produce dense point clouds with larger coverage but lower accuracy. LIDARs, on the other hand, have higher accuracy and longer range but much less coverage. LIDARs are also more expensive. The research question examines whether incorporating LIDARs can significantly improve stereo camera accuracy. Current sensor fusion methods use LIDARs’ raw measurements directly; thus, the improvement in estimation accuracy is limited to only LIDAR-scanned locations The main contribution of our new method is to construct a reference ground plane through the interpolation of LIDAR data so that the interpolated maps have similar coverage as the stereo camera’s point cloud. The interpolated maps are fused with the stereo camera point cloud via Kalman filters to improve a larger section of the topography map. The method is tested in three environments: controlled indoor, semi-controlled outdoor, and unstructured terrain. Compared to the existing method without LIDAR interpolation, the proposed approach reduces average error by 40% in the controlled environment and 67% in the semi-controlled environment, while maintaining large coverage. The unstructured environment evaluation confirms its corrective impact.

## 1. Introduction

Topography estimation is vital for autonomous navigation in off-road terrain environments because the main challenge for terrain navigation comes from static obstacles and impassable terrain. These challenges are shared by several application domains where automation is desired in unstructured outdoor environments, such as agriculture, mining, and forestry [1,2,3]. Additionally, they are relevant to search and rescue operations and disaster response efforts, where the environments are harsh and unpredictable.

There are many methods of estimating the topography and traversability of terrain areas. The fundamental method is a two-dimensional (2D) occupancy grid or traversability map. A 2D grid map partitions the ground and each grid contains a binary variable indicating whether an obstacle occupies it. A vehicle can pass the grid if and only if the grid is not occupied. Sock et al. [4] fuse measurement from a LIDAR and a camera to estimate the ground geometry and determine the local traversability of every grid. This method, however, loses the precise height information about the terrain. Although three-dimensional (3D) occupancy grids [5] give a rough estimate of the height, the precision is limited by the large size of the grid. To precisely model the terrain height, Fankhauser et al. [6,7] use a pseudo-3D (2.5D) grid representation of the terrain where 2D grids still partition the ground but each grid cell contains a continuous variable to represent the average terrain height of the grid. This method is also called elevation mapping, which has the benefit of keeping the shape of the terrain.

All these examples are for navigation applications but topography estimation is also applied for general mapping such as for the forest industry to check both the inventory of trees as well as to create elevation maps for future planning of harvesting, planting, and other operations [8,9]. Efforts have also been made to measure and map snow depth [10], as well as to support archaeological research, such as the mapping of Mayan cities, made possible by the Pacunam LIDAR Initiative [11].

Most topography estimation methods work on point cloud data provided by either image or Time of Flight (TOF) sensors. The most common ones are LIDARs and stereo cameras but mono cameras or radars may also work [12]. The benefit of using stereo cameras and LIDARs is their convenience of quickly capturing enough 3D point cloud data for representing the environment. The selection between stereo cameras and LIDARs is an important decision. The advantage of a stereo camera is the high resolution that can generate a very dense point cloud [13,14], but its disadvantage is the sensitivity to light saturation, which generates faulty measurement points. By contrast, LIDARs have both a larger measurement range and higher accuracy than the stereo camera but have much smaller coverage of the ground than the stereo camera. The measurement accuracy of the LIDAR is also inversely proportional to the distance. Moreover, LIDARs, especially 3D LIDARs, are more expensive than stereo cameras.

As presented in a survey published by Bai et al. [15], and by specific agricultural and planetary applications [16,17], a stereo camera alone is sufficient for some autonomous navigation machines. However, more popular perception systems for autonomous driving use LIDARs or combine stereo cameras and LIDARs. Given the much higher cost of LIDARs, a fundamental question is whether an additional LIDAR can significantly increase a stereo camera’s accuracy and reliability. A common argument for combining a LIDAR and a stereo camera is that the camera has high resolution in the short to medium range (1–15 m) and the LIDAR improves the measurement of the camera and increases the range up to 90 m.

Traditional methods fuse the two sensors’ data by combining their measurement data and variances at locations where both sensors have measurement data. A limitation of these existing fusion methods is that the sparse LIDAR data can only improve the measurement accuracy of a small percentage of the stereo camera data. The main contribution of this paper is to construct a reference ground plane through interpolation of LIDAR data so that the interpolated maps have similar coverage as the stereo camera’s point cloud measurement. This ground plane includes both a height map and a variance map, with variance increasing as the distance from LIDAR measurements grows. The interpolated maps are fused with the stereo camera point cloud via Kalman filters to improve a larger section of the topography map.

This paper is organized as follows: Section 2 reviews related work on existing methods for topography estimation using point clouds as input. It also explores techniques for fusing data from different sensor types, particularly LIDAR and stereo cameras. Section 3 explains the approach for estimating the measurement variance of LIDAR and stereo cameras, which will later inform the fusion of different point clouds discussed in Section 4, where the overall methodology of this work is also presented. Section 5 outlines the results, beginning with controlled lab experiments on a known topography, followed by tests conducted in a more unstructured environment. Finally, Section 6 provides a discussion of the results and presents the conclusions.

## 2. Related Work

Topography estimation can be conducted with many types of range sensors. The most popular options are LIDAR and stereo camera. Both generate a large number of data as point clouds. LIDARs are preferred since they are more precise but the disadvantage is the sparsity of the point cloud [13,18]. LIDARs generate a set of lines of measurement points, resulting in dense measurement data along the lines but no data points between the lines. Stereo cameras have less range and less precision. The measurement errors become larger if the measurement points are further away from the camera [19,20] or have discontinuous depths [21,22]. The advantage of the stereo camera is to generate a much more dense point cloud that evenly covers the whole range. Usually, the LIDAR’s sparseness is not a big problem, because it can scan the environment over time when it is moving [23]. The main problem of this approach is that any changes in the environment are only updated while a line scans the terrain and then the environment is assumed to be static until another line hits the area again. It would be preferable to collect range data for the whole environment at each sensor reading but that is not possible with LIDAR. A solution to the problem is to combine the LIDAR data with stereo camera data to cover the whole area at each update [24,25,26].

While stereo cameras can measure a large area and generate a dense point cloud, they are sensitive to lighting conditions, which affects the variance of their measurements [13]. Range measurements from stereo cameras are realized by matching areas in the pictures from the two cameras and then calculating the distance of that area from the sensors. Since matching requires identifiable areas in both pictures, the method is sensitive to light saturation and homogeneous or texture-less regions in the images [19,20]. In these situations, the stereo camera cannot match areas between the two pictures [22].

The LIDAR data could alleviate this issue if used as a high-precision reference for the image data. There are several approaches to merging data from the two sensors. A common approach is to merge the LIDAR data into the stereo camera image by either interpolating the LIDAR data between its lines [27,28] or using other types of interpolation as in [29]. The goal of these approaches is to enhance the information in the images through improving the distance estimation to objects or segmented areas. Wolf and Berns [24] insert the stereo camera data into the measurement gap of the sparse LIDAR point cloud while also filtering the stereo camera points based on the variance in the stereo camera points. The variance in the stereo camera’s point cloud is inversely proportional to its distance from the camera. This reduces the measurement range of the stereo camera. There are other ways of handling the difference in data quality as presented in [6], where a common map is updated with both sensors’ values and variances. This results in a map that stores the mean and variances of height values. This allows the map to use all available sensor data while maintaining the data’s quality. A drawback of the typical map approach is that certain regions only have low-quality stereo camera data due to the sparseness of the LIDAR data. Over time this is handled while moving since the LIDAR will be able to cover the uncertain areas, but each update will have the same issue due to the sparseness of the LIDAR.

One way to alleviate the issue with sparse measurement data is to merge the data into a persistent map as conducted by [6,30]. The methodology is based on having a persistent map centered on the “actor” which updates based on the actor’s movement and new sensor data. Fankhause et al. [6] update a local robot-centric map based on both the sensor data and information about the movement of the actor. Pan et al. [30] split the area around the “actor” into a grid and then updated the grid based on point cloud data and the variance. The method creates a persistent global map. This can be used to fuse data from different types of sensors based on their variance and allows LIDARs to scan the environment over time. This works well as long as not too much of the close environment is occluded. An example of problematic occlusion is when an actor climbs an uphill, the slope occludes the other side of the slope. When the actor reaches the top, historical data for the area in front of the actor are missing. In that situation, the topological map in front of the actor can only be generated from the current sensor data. By contrast, if the actor uses a combination of LIDAR and stereo camera, it must mostly rely on low-quality stereo camera data because the LIDAR has not yet scanned most of the area. Efforts have been made to address the resulting gaps using machine learning and AI techniques [31,32,33], where various types of neural networks are employed to fill in missing sensor data. As shown in [32] the network successfully fills in the data when the environment is even and struggles more when there are more obstacles around. This may result from how the network is trained; however, when the environment becomes more unstructured, fewer patterns can be used to predict the shape of the terrain. This makes it even more important to exploit the recorded sensor data due to the difficulty of reconstructing it.

While the persistent map improves the sensor fusion of stereo cameras and LIDARs, it inherits the same problems of the measurement data of the two types of sensors. A solution is to create an interpolation map from the LIDAR data [27,28] and then use the interpolated data as a rough reference map to improve the measurement data from the stereo camera.

## 3. Prerequisite

### 3.1. LIDARs

There are two main types of LIDARs: mechanical and solid-state. Mechanical LIDARs have either a single or an array of emitter and receiver modules to generate up to $360^{\circ }$ view around the sensor. Solid-state LIDARs have no moving parts and use a solid-state laser and a photodetector to measure distances in a specific direction. Solid-state LIDARs are less prone to failure but generally have a smaller field of view and less range than mechanical LIDARs. LIDARS usually describe their error as static in their datasheets due to how low the error generally is over its range. However, this error is an average over the whole span and thus does not capture how accuracy is reduced over that range. For example, the VLP-16 LIDAR is stated to have an error of ±3 cm [34]. For either LIDAR type the method of measuring distances is the same, using laser TOF measurements to find the distance to an object. Thus, the deviation for both types can be estimated by looking at how the laser beam widens with increased distance from the emitter. If the beam is modeled like a cone as conducted by [24]. The width of the beam can be viewed in Figure 1. Where the original radius of the beam is half the optics coverage diameter *c* and the increase in diameter is decided by the beam distribution factor *b* multiplied by the distance from the sensor emission point. Which lets the resulting deviation be calculated as in Equation (Equation 1). This assumes that the angle of the emission factor is small, which for lasers holds. The parameters for a SICK LMS 511 are c=0.014 m and b=0.0119 rad.(1)$$\sigma_{l}=l\cdot b+\frac{c}{2}$$

### 3.2. Stereo Cameras

Stereo cameras estimate depth by triangulating the position of objects using the disparity between two mono-cameras. Stereo cameras inherit disturbances affecting mono-cameras and the disparity matching that generates the depth measurements. This results in problems for measurement points close to the camera’s maximum and minimum range due to increased distortion with increased distance. This has led to experimentally derived deviation models for estimating depth deviation. Such a model is presented by Ortiz et al. [35], and shown in Equation (Equation 2). Where a,b are the exponential parameters, *l* is the distance in meters from the camera and $\sigma_{c,base}$ is the starting deviation. These parameters are set experimentally and for a ZED camera at resolution 1920 × 1080, the parameters have the following values; $\sigma_{c,base}=0.06,a=0.0106,b=0.2215$ [35].(2)$$\sigma_{c}=\sigma_{c,base}+a\cdot e^{b\cdot l}$$

Other work has shown that the deviation also can be modeled as a quadratic approximation [36], in this case for the RealSense D435 with the resolution 848 × 480. The approximation is shown in Equation (Equation 3) with parameters $\sigma_{c,base}=0.0012,a=0.0007,b=0.0039$ and *l* as the distance from the camera in meters.(3)$$\sigma_{c}=\sigma_{c,base}+a\cdot l+b\cdot l^{2}$$

From the deviation models, it is clear that deviation for stereo cameras increases much faster with distance with either an exponential or quadratic relationship while the increase is linear for LIDAR. That linear relationship is barely considered and a static measurement is more commonly used to describe the accuracy of LIDARs.

## 4. Sensor Fusion Methods

This section examines the impact of two types of 3D scanning sensors, LIDAR and stereo camera, as well as their combination for topography estimation. To ensure thorough evaluation, both sensors were used to capture terrain data in a controlled lab setting, a semi-controlled outdoor environment, and a natural forest setting. The sensors were positioned statically, and raw data were collected in the form of point clouds. A voxel filter then filtered these point clouds [37], and the filtered point clouds were used to generate a model of the terrain.

The data from each sensor are used to generate a separate terrain model, which is compared against ground truth measurements. Then, the sensor data are fused to generate a combined model to determine whether using both sensors improves the situation.

The depth sensors used for data collection were a Velodyne VLP-16 [34] and a ZED-2 stereo camera [38]. The Velodyne is a 16-line 360° 3D LIDAR and has been a common choice for robotic applications, for example, by ANYbotics. The ZED-2 is a stereo camera that also includes internal perception sensors such as IMU and barometer. The largest difference between the two different depth sensors is the field of view, range, and point cloud density. The stereo camera has a much smaller field of view of 120° and a range of 15 m while the LIDAR can measure up to 100 m with a 360° field of view. On the other hand, the stereo camera can generate a much denser point cloud. To make a fair comparison possible, both sensors will be compared within the range of the stereo camera but the base differences in the behavior of the sensors should be taken into account in the overall evaluation.

The mounting of the sensors is shown in Figure 2a. The goal with the mounting locations was to keep the sensor mounted as close as possible to a single point in space to simplify the fusion of sensor data even though this resulted in mounting the stereo camera upside down. This setup allows one to compare data collected from both sensors simultaneously. To interface with the sensors, the Robot Operating System (ROS) [39] is used as middleware since both sensors have readily available software packages that convert their raw measurements into point clouds; the ROS setup is shown in Figure 2b. These point clouds are sent as timestamped messages over the ROS network. If the relative positions of the sensors are specified as static transformations it becomes trivial to match both point clouds in space. These timestamped point cloud messages and their static transformations are stored in a “rosbag” (a file format designed for recording and replaying messages such as sensor data) which can later be used for comparison and topography estimation. The topography estimation algorithms are implemented by MATLAB [24.2.0.2712019 (R2024b)] in a laptop PC.

### 4.1. Baseline (Naive) Approach

The baseline or “Naive” approach is implemented using a similar approach as in [6], where an elevation map is made by transforming position coordinates and measurement variances of point cloud data from the sensor frame to the grid map frame. The points in 3D space are projected onto a 2D grid, where (x,y) defines the location of the measurement and $\tilde{p}$ is the height measurement. The measurements are estimated by a Gaussian probability distribution with mean $p=\tilde{p}$ and variance $\sigma_{m}^{2}$ calculated as in Equation (Equation 4), where $J_{S}$ is the Jacobian for the sensor measurement, $J_{\Phi }$ is the Jacobian for the sensor frame rotation. These Jacobians describe the mapping from a three-dimensional measurement in the sensor frame to a scalar height measurement in the map frame. The sensor frame is assumed to have a known static transform to the map which means it is possible to transform measurements freely between the two frames. $\Sigma_{S}$ is the covariance matrix of the sensor model and its values are decided from range sensor noise models as mentioned in [6,40]. $\Sigma_{\Phi }$ is the covariance matrix for the sensor rotation which is ignored like in [6] due to how the map frame is defined for both their and our method.(4)$$\sigma_{m}^{2}=J_{S}\Sigma_{S}J_{S}^{⊺}+J_{\Phi }\Sigma_{\Phi }J_{\Phi }^{⊺}$$

The elevation map is a 2D stochastic function that maps a 2D coordinate (x,y) to the corresponding height estimation that satisfies Gaussian distribution$$M\left(x,y\right)∼N(h\left(x,y\right),\sigma_{h}^{2}\left(x,y\right))$$
The height estimation is iteratively refined by fusing the measurement data of a LIDAR and a stereo camera. The height measurements at grid (x,y) of the LIDAR and the stereo camera are both Gaussian variables$$m_{l}\left(x,y\right)∼N\left({\tilde{p}}_{l}\left(x,y\right),\sigma_{l}^{2}\left(x,y\right)\right)$$$$m_{c}\left(x,y\right)∼N\left({\tilde{p}}_{c}\left(x,y\right),\sigma_{c}^{2}\left(x,y\right)\right)$$

The subscript *l* or *c* represents the LIDAR or the stereo camera. Since the two sensors have different update frequencies, the height estimation at a grid (x,y) is updated once measurement data from either sensor are obtained. The updating algorithm applies a one-dimensional Kalman filter at every grid. For brevity, the following formulas for the 1D Kalman filter algorithm skip the grid coordinate.

The state update and sensor measurement model for the Kalman filter algorithm is given by Equation (5). Since the elevation map is assumed static and the update process has no disturbance, the height prediction $h_{k}$ at step *k* is identical to the estimate at the last step k−1. The height sensor has Gaussian noise $n_{k}∼N\left(0,\sigma_{m}^{2}\right)$. If at step *k*, the available measurement data are from the LIDAR, then ${\tilde{p}}_{k}={\tilde{p}}_{l,k}$ and $\sigma_{m}^{2}=\sigma_{l}^{2}$; otherwise, ${\tilde{p}}_{k}={\tilde{p}}_{c,k}$ and $\sigma_{m}^{2}=\sigma_{c}^{2}$.(5a)$$\begin{array}{cc} h_{k} & =h_{k-1} \end{array}$$(5b)$$\begin{array}{cc} {\tilde{p}}_{k} & =h_{k}+n_{k} \end{array}$$

Suppose that the height estimate at step k−1 has mean ${\hat{h}}_{k-1}$ and variance $\sigma_{h,k-1}^{2}$. The initial estimates of ${\hat{h}}_{0}$ and $\sigma_{h,0}^{2}$ are determined by the first sensor measurement. For any k≥1, the a priori prediction of height at step *k* has the following mean and variance(6)$$\begin{array}{cc} h_{k|k-1} & ={\hat{h}}_{k-1} \end{array}$$(7)$$\begin{array}{cc} \sigma_{h,k|k-1}^{2} & =\sigma_{h,k-1}^{2} \end{array}$$
The innovation variance and the Kalman gain for this problem are$$S_{k}=\sigma_{h,k-1}^{2}+\sigma_{m}^{2},\;K_{k}=\frac{\sigma_{h,k-1}^{2}}{\sigma_{h,k-1}^{2}+\sigma_{m}^{2}}$$
Therefore, the mean and variance of the a posteriori height estimate are(8a)$$\begin{array}{cc} {\hat{h}}_{k} & =h_{k|k-1}+K_{k}\left({\tilde{p}}_{k}-h_{k|k-1}\right)=\frac{\sigma_{h,k-1}^{2}{\tilde{p}}_{k}+\sigma_{m}^{2}{\hat{h}}_{k-1}}{\sigma_{h,k-1}^{2}+\sigma_{m}^{2}} \end{array}$$(8b)$$\begin{array}{cc} \sigma_{h,k}^{2} & =\sigma_{h,k|k-1}^{2}-K_{k}S_{k}K_{k}^{T}=\frac{\sigma_{h,k-1}^{2}\sigma_{m}^{2}}{\sigma_{h,k-1}^{2}+\sigma_{m}^{2}} \end{array}$$

This update process adjusts the map according to the confidence in both the measurement and the existing map data. It works well when the terrain is smooth but struggles when there are large vertical obstacles such as trees and steep rock surfaces. To handle these cases, we use the $\chi^{2}$ test to judge whether a new estimation by Equation (8) is acceptable [30]. The $\chi^{2}$ value is computed by Equation (Equation 9).(9)$$\Delta =\frac{|\hat{h_{k}}-{\tilde{p}}_{k}|}{\sigma_{h,k}^{2}}$$

The threshold value of the $\chi^{2}$ test is 3.84 as there is one degree of freedom and we use a *p*-value of 0.05. If $\chi^{2}\leq 3.84$, the new estimation value ${\tilde{p}}_{k}$ passes the test, and the elevation map is updated by Equation (8). If $\chi^{2}>3.84$, the new estimation value does not pass the test and the elevation map is updated by Equation (Equation 10).(10)$${\hat{h}}_{k}=max\left({\tilde{p}}_{k},{\hat{h}}_{k-1}\right)$$

Equation (Equation 10) defines two cases. Case 1, if the current height estimate is larger than the last height estimate, i.e., ${\tilde{p}}_{k}\geq {\hat{h}}_{k-1}$, then ${\hat{h}}_{k}={\tilde{p}}_{k}$ and $\sigma_{h,k}^{2}=\sigma_{m}^{2}$. Case 2, if the current height estimate is smaller than the last, i.e., ${\tilde{p}}_{k}<{\hat{h}}_{k-1}$, then ${\hat{h}}_{k}={\hat{h}}_{k-1}$ and $\sigma_{h,k}^{2}=\sigma_{h,k-1}^{2}$. This results in an update process as shown in Figure 3a.

The update process can be extended to fuse measurements from different sources into the same map by letting $\sigma_{m}^{2}$ depend on the source of the measurement. In our case that would result in two different measurements; $m_{l}$ and $m_{c}$, where index *c* corresponds to the camera and *l* to the LIDAR. Let both updates go through the check-in Equation (Equation 9). The update process is shown in Figure 3b.

The approach can be viewed as a basic approach or “Naive” since it considers only precision and only uses the raw measurement points from both sensors. Besides precision, there are other large differences between stereo cameras and LIDARs. LIDARs are very precise but measure only a set of lines around the sensors, in the case of VLP-16 it has 16 lines, which will result in a much smaller density of points in the point cloud. Stereo cameras have a much larger density of points dependent on the resolution of the cameras but less precision for each point. Just combining the two using Equation (8) will increase the precision for the locations that are covered by both stereo cameras and LIDARs. The precision of the locations without LIDAR measurement remains unchanged. The discrepancies between the raw data from the two sensors, particularly the variations in point density and the stereo camera’s inaccuracies (such as detecting negative height values where none exist, especially in the lower-right of the point cloud) are illustrated in Figure 4a,b. The ground truth of the measurements is shown in Figure 4c. The ground truth is a flat area with a size of 7.5 m × 10 m with a map resolution of 0.1 m. The area contains four blocks whose heights and locations are the following: 0.09 m at (57, 46), 0.16 m at (51, 60), 0.43 m at (61, 58), and 0.57 m at (65, 43). The position of a block is determined by its bottom-left corner.

This key difference in the point cloud will restrict the extent to which LIDAR can enhance the elevation estimate during updates. To improve the accuracy of the elevation map the LIDAR measurements can be interpolated to create a rough estimate of the terrain for improving the quality of the stereo camera data. This results in an interpolation-based approach.

### 4.2. Interpolation Based Approach

The interpolation-based approach requires two different update processes. First, when a LIDAR point cloud is received, it is used to both update a combined map and a separate LIDAR map. The LIDAR map is linearly interpolated between the LIDAR lines to cover the area. The interpolation is performed along the depth in the sensor’s perspective, specifically in the x-direction as illustrated in Figure 5a. The rows are iterated for each column until two distinct measurements are detected. The cells between the two points are then estimated with height values using linear interpolation of the measured values. For example, the cells between $p_{1}$ and $p_{2}$ in Figure 5b are computed based on the height values at those two points. The column is considered complete once the iterator moves past $p_{2}$ with no further data points to interpolate. The process then proceeds to the next column, where the cells between $p_{3}$ and $p_{4}$ are filled similarly.

For the interpolation, the map can be viewed as a function that takes a coordinate pair $p=(x_{p},y_{p})$ and returns the height at that location as described in Equation (Equation 11). The interpolation is conducted between two known points in the same column, with the same y-value, as shown in Equation (12). The height value of a location, $x_{p}$, is calculated using the height values at adjacent measurement points with the same *y* coordinate as $y_{p}$. Denote the *x* coordinates of the two adjacent points as $x_{0}$ and $x_{1}$. It must be true that $x_{0}<x_{p}<x_{1}$. Denote the measured height values at the two adjacent points $\left(x_{0},y_{p}\right)$ and $\left(x_{1},y_{p}\right)$ as $h_{0}$ and $h_{1}$, respectively. The height at the known location is excluded from the interpolation calculation to avoid updating from the same point cloud data twice.(11)$$M_{h}\left(x_{p},y_{p}\right)=h_{p}$$(12a)$$h_{p}=k\cdot \left(x_{p}-x_{0}\right)+h_{0}$$(12b)$$k=\frac{h_{1}-h_{0}}{x_{1}-x_{0}}$$

This creates a rough reference plane for the true plane but it does not consider the uncertainty that is inherent to interpolating between known values. The uncertainty of the interpolated values should be low at locations close to the measured points and become larger when the locations are further away. To capture the uncertainty of the interpolated height values, an uncertainty/variance map of the height values is created alongside the interpolated height. The uncertainty map matches the interpolated height map but contains the variance of the height values as defined in Equation (Equation 13).(13)$$M_{\sigma }\left(x,y\right)=\sigma_{p}^{2}$$

The variances for the interpolated height values are calculated as a ramp that increases with the distance from a measured point until the variance reaches a maximum value according to Equation (14). The ramp constant $c_{\sigma }$ as shown in Equation ([Disp-formula FD14a-sensors-25-00509]) is calculated based on the variance of the sensor $\sigma_{s}^{2}$, the maximum variance $\sigma_{max}^{2}$ and the distance where the variance reaches its maximum value $d_{max}$. Using $c_{\sigma }$ the variance in each cell between two measure points is calculated with Equation ([Disp-formula FD14b-sensors-25-00509]), where $d_{1}$ and $d_{2}$ are the distances to the measured points as shown in Figure 5b. Before saving the variance in the map, the variance is saturated by $\sigma_{max}^{2}$ as in Equation ([Disp-formula FD14c-sensors-25-00509]).(14a)$$c_{\sigma }=\frac{\sigma_{max}^{2}-\sigma_{s}^{2}}{\sigma_{s}^{2}\cdot d_{max}}$$(14b)$$\sigma_{p,inter}^{2}=\sigma_{s}^{2}+\sigma_{s}^{2}\cdot c_{\sigma }\cdot \min\left[d_{1},d_{2}\right]$$(14c)$$\sigma_{p}^{2}=\min\left[\sigma_{p,inter}^{2},\sigma_{max}^{2}\right]$$

The resulting variance function between two measured points becomes a trapezoid between two measured points where the variance increases as a ramp until distance $d_{max}$ from one point where the variance has reached $\sigma_{max}^{2}$ which it keeps until it ramps down as the distance to the next measured point is less than $d_{max}$ as shown in Figure 5b.

The function can be modified by changing the design parameters $d_{max}$ and $\sigma_{max}^{2}$ to control the ramp and the height of the trapezoid. A steeper slope means that weaker assumptions are made about the areas around the measured points and a lower $\sigma_{max}^{2}$ means that the interpolated values in Equation (Equation 11) are allowed to affect the combined height map more. An example of an interpolated map generated from the LIDAR data in Figure 4b is shown in Figure 6a with the corresponding variances with design parameters $d_{max}=0.5$ m and $\sigma_{max}^{2}=0.99$ shown in Figure 6b. Note that the regions containing sensor data are left empty.

These maps are used as a reference while updating the combined elevation map from the more unreliable stereo camera point cloud. The update procedure is outlined in Algorithm 1. In this algorithm, the variable “pointCloud” represents a point cloud structured as a multidimensional array, i.e., $\mathrm{pointCloud}\in R^{3\times n}$, with *n* denoting the number of points. The different maps: $M_{in}$, $M_{out}$, $M_{inter}$ are all structures that contain a pair of 2D matrices of the same dimensions. For instance, $M_{inter}$ contains both the interpolated height map $M_{inter,h}$ and the interpolated variance map $M_{inter,\sigma }$ as described in Equations (Equation 11) and (Equation 13) (see Line 1).

Upon receiving a point cloud, the algorithm first determines its source. If the point cloud is from the LIDAR, $M_{inter}$ is re-initialized (at Line 3) via the **init** function, which clears the two sub-maps. The algorithm then updates the combined map with sensor data. The **findMapIndex** function determines the appropriate index for each point (at Line 5), and the **fusePoint** function integrates the measurement into the existing map $M_{in}$, resulting in an updated map $M_{in}$ (at Line 6). Subsequently, the interpolation map $M_{inter}$ is updated and interpolated using the **interpolate** function, applying Equations (12) and (14) (at Line 9).

The process follows a similar pattern for point clouds originating from a stereo camera. After integrating each point into the combined map (at Line 13), the algorithm verifies if a corresponding height value exists in the interpolation map (at Line 14). If a height value is present, the interpolated value is combined with the stereo camera point in the map. Once all points have been integrated, the algorithm produces the updated map (at Lines 19–20).

While this study does not specifically address the implications of real-time online implementation, we qualitatively analyze the computational complexity of the algorithm here. The primary computational challenge lies in fusing the stereo camera data into the common map due to the density of the point clouds. In contrast, the additional task of interpolating LIDAR data (at Line 9) is considered negligible. While the fusion of the interpolated map (at Line 15) slightly increases the processing time for stereo camera data, it eliminates the need to determine precise positions for the interpolated data points. As a result, it mainly adds a simpler fusion calculation, which has a smaller impact compared to the steps involved in stereo camera point fusion (in Lines 12–13). Additionally, since off-road machinery generally operates at slower speeds, more time is available for these calculations, reducing the demand for high update rates and minimizing the impact of sensor movement. This movement can be further addressed by employing dynamic map update methods based on sensor motion [6].
**Algorithm 1:** Fusion Algorithm.
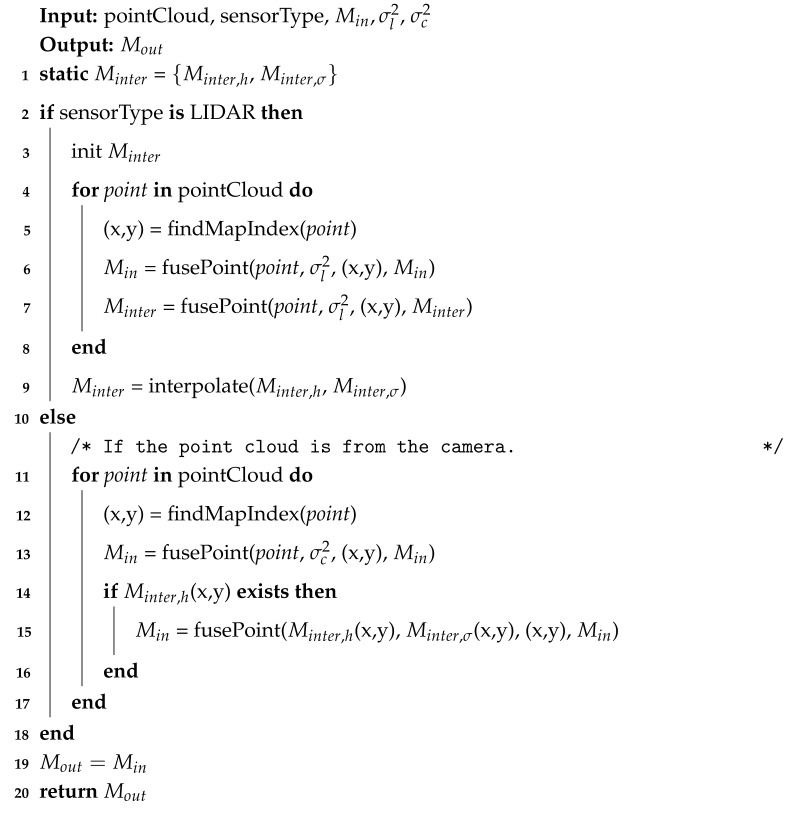


## 5. Results

The proposed topography estimation method is tested in three environments: a controlled lab environment, a semi-controlled outdoor environment, and an unstructured forest environment. The lighting conditions for all environments are selected to have a small influence on the stereo camera. In the controlled environment, a small area that reflects light to the camera causes a larger error in the stereo camera, but the error is corrected by LIDAR using our method. In the semi-controlled environment, the site is shaded by a building and hence has sufficient and uniform lighting. In the unstructured environment both shade from surrounding trees and overcast weather result in even lighting conditions favorable for the stereo camera. Examples of point clouds, single sensor maps, and fused maps are presented for all environments. In Figure 7, Figure 8, Figure 9, Figure 10, Figure 11, Figure 12, Figure 13, Figure 14, Figure 15, Figure 16, Figure 17 and Figure 18 below, when X-Y frames and color coding are applied, X and Y values represent the discrete indices of the ground map, and the color indicates either the height or the corresponding variance value.

### 5.1. Controlled Lab Environment

The controlled environment with ground truth height values is shown in Figure 4c. The following experiments and evaluations compare the maps generated from either stereo camera or LIDAR measurements to the ground truth map, followed by the results from the two fusion methods compared to the ground truth.

#### 5.1.1. Stereo Camera

Creating the height map with only the stereo camera’s range data results in a dense map but errors in some areas are very large. This occurs due to homogeneous regions (such as flat, single-color areas or regions with light saturation) and obscured areas. The algorithm used to calculate the distance using stereo images prefers continuous surfaces. When objects obstruct parts of the scene, the algorithm often “smears” the range data to create a continuous surface. Non-uniform conditions also increase the variance in the map which already has a larger baseline variance due to the sensor type as shown in Figure 7b. Examples of large errors due to homogeneous and obscured areas can be seen at (35, 72) and (60, 25) in Figure 7a and Figure 8a where the stereo camera suddenly measures negative values where the region should be flat. An example of the smearing can be seen around (60, 65) in Figure 7a and Figure 8a where instead of detecting the end of the block, the height values behind the block are estimated as the same or similar to the height at the block into the background. The error of the stereo camera map compared to the ground truth map is shown in Figure 8a.

#### 5.1.2. LIDAR

Maps generated from only LIDAR data are more precise and have no smearing and light sensitivity but they are sparse due to the sensors being stationary as seen in Figure 7c,d. The error from the ground truth is presented in Figure 8b. There is an option to interpolate the LIDAR data to fill the map but this results in precise sections along the LIDAR lines and very imprecise areas between the lines, as shown in Figure 6, where the height between the LIDAR measurements is calculated as a linear slope. This can most clearly be seen around (86,64) in Figure 6a. This is similar to the smearing problem with the stereo camera maps since it creates the same problem with finding discontinuous surfaces. This would be less of an issue along continuous surfaces, like for the stereo camera, but reduces the precision for discontinuous surfaces.

#### 5.1.3. Naive Combined Map

As a baseline, the data from the two sensor types are combined straightforwardly using the update functions outlined in Equation (8). In this method, the combined map is updated with data from both sensors without applying any interpolation. Overall, the resulting map typically shows reduced error compared to the map created using the stereo camera. The reason is that the error is decreased where the LIDAR data are available. The LIDAR data also help to counteract some of the smearing effects. It is possible to see the lines from the LIDAR on the right-hand side of Figure 9a and they are even clearer in the error map, shown in Figure 10a, where the error is locally reduced where there are LIDAR data. A similar effect on the map variance can be observed in Figure 9b where a reduction in variance along the LIDAR lines is evident compared to the stereo camera map variance.

#### 5.1.4. Interpolation Based Combination

When interpolation of the LIDAR data are added, the error throughout the map is reduced. More importantly, the areas with large errors from the stereo camera data (around (35, 72) and (60, 25)) are greatly reduced. It is now possible to see the larger boxes from the ground truth in the map as seen in Figure 10b where the error is low. The errors resulting from the smearing in the stereo camera data are also reduced in non-obscured areas but not for areas that are obscured from both sensors as seen in the area around (60, 65) in Figure 9c. An effect of the higher variance in the interpolated height values can also be observed in regions where only stereo camera data are available, as shown in Figure 9d.

The errors and fill percentage of each method are shown in Table 1. From a precision perspective, the LIDAR map outperforms each of the other types but it covers a much smaller area. The LIDAR map covers a fourth of the other map areas, which further shows the problem with only using a LIDAR; it provides precise measurements but its coverage is low. Even with that in mind, the interpolation-based map gets close in precision in terms of mean error and root mean square error.

Compared to the map constructed from only the stereo camera, both the naive and the interpolation methods have improvements. The naive approach improves the map slightly since it uses limited LIDAR data. The interpolation approach estimates the heights of gaps between LIDAR measurement lines and achieves larger improvement. Compared to the naive approach, the interpolation approach reduces both mean error and RMSE by around 40% but has less reduction on max error. This is because there are little LIDAR data at the locations where the stereo camera has the largest error.

### 5.2. Semi-Controlled Outdoor Environment

To show the robustness of the proposed method, the method was further evaluated outdoors in a semi-structured setting on uneven asphalt, with objects of known dimensions placed at specified locations, as shown in Figure 11. The heights of the marked points in the photo were measured by a ruler and their heights are presented in Table 2. Examples of the resulting point clouds are presented in Figure 12.

The single sensor maps are shown in Figure 13. In this new environment, the stereo camera could not cover the entire area, leading to large gaps in the point cloud, particularly around objects. This issue is due to the abrupt distance changes caused by the objects. Additionally, lighting conditions result in an uneven terrain representation compared to the LIDAR data. The stereo camera also inaccurately placed objects 4 and 5 at (57, 40) to an incorrect position at (72, 33). While the LIDAR correctly estimated the object’s height as 45 cm, the stereo camera measured it as 37 cm. For objects on the left side, both LIDAR and the stereo camera return to the same position, but the stereo camera’s map shows them smeared into the background. The stereo camera estimated their height above ground as 19 cm, while the actual height is 27 cm, closer to the LIDAR’s measurement of 25 cm.

When fusing sensor measurements using both the naive and interpolation methods, the distinct characteristics of the two sensors become apparent, particularly in the naive fusion, as illustrated in Figure 14. While measurements are relatively similar close to the sensors, discrepancies grow with distance from them. The largest differences occur around objects and near the concrete wall at the far end of the map. In the naive map, discrepancies are limited to the areas along LIDAR lines, whereas the interpolated map exhibits larger differences in the central and left-side regions of the map.

For both fusion methods, the map accuracy improves compared to the stereo camera alone. Since we have ground-truth height values only at the five marked points in Figure 11, we compare the elevation maps by the naive and the interpolation methods only at the five points. These results are presented in Table 3. The stereo camera map exhibits the highest error, while the LIDAR map shows the lowest, as expected. The primary difference between the naive and interpolation maps is observed in the maximum error. However, the interpolation method consistently outperforms the naive approach across all metrics, achieving a 67% reduction in mean error and a 71% reduction in RMSE.

### 5.3. Unstructured Forest Terrain

The last case study measures the elevation map of an unstructured forest terrain with trees and grasses. Figure 15 is a photo of the test environment. Figure 16 shows the measurement point clouds by a LIDAR and a stereo camera.

When creating the elevation map, we ignore all points higher than a threshold to filter away branches and leaves, because the focus of the map is on the ground topography. As shown in Figure 16 while there is more coverage by the stereo camera point cloud, it is also more noisy. The stereo camera can handle continuous surfaces reasonably well but has issues with discrete changes such as those around the tree trunks that look distorted and offset compared to the LIDAR point cloud. This happens due to smudging because the stereo camera attempts to generate a continuous surface across the point cloud data even across discontinuous objects, which distorts sharp edges. The positional difference for the right tree is significant. For example, the highlighted tree in the LIDAR data, located at (31, 26), is shifted to (40, 25) in the stereo data.

The average of the stereo camera points is close to the LIDAR points, but there is more noise in the stereo camera measurements. It is clear that while a stereo camera could be used to create a semi-accurate ground model it has clear issues to capture harsh changes in the topography. The resulting single-sensor maps derived from the point clouds are displayed in Figure 17. These maps illustrate that while the stereo camera map covers a larger portion of the area, its measurements are noisier, as indicated by the higher variance shown in Figure 17b.

When the data from both sensors are fused using Equation (8), the LIDAR can correctly add the positions of the trees using either the naive approach or the interpolation method as shown in Figure 18. For both approaches, erroneous positioned trees are still there. A possible reason for this error is that some leaves or branches are present at the position and they are mistakenly identified as a tree. The LIDAR lines are also visible in the naive map as they were for the tests in a controlled environment presented in Figure 9a. They are mostly visible in the area to the center-left at approximately (42, 32), where the LIDAR is assumed to find the correct height. If this specific area is compared to the stereo camera map, Figure 17a, the ground is probably close to a linear slope. With these assumptions, the interpolation should help correct the stereo camera data in this area. Figure 19 presents a cross-section along Y = 32. The results from the naive estimation method, shown in red, display sharp edges where the LIDAR corrects the estimates at X = 32, X = 43, and X = 59. In contrast, the interpolation method, depicted in blue, extends this influence over a wider area, gradually distributing the correction.

## 6. Conclusions

A new method utilizing a LIDAR-based reference ground plane has been developed to address stereo camera inaccuracies in topography mapping. The new method constructs a reference ground plane with similar coverage as the stereo camera’s point cloud through interpolation of LIDAR data. The interpolated maps are fused with the stereo camera point cloud via Kalman filters to improve a larger section of the topography map. The process operates in two parts: the first part happens when LIDAR data are received. The data are used to update the fused topography estimation map and create the interpolated reference maps. Note that both maps contain the estimated height matrices and the variance matrices. The second part happens when stereo camera measurements are received. Then the fused topography map is updated by incorporating both the stereo camera data and the previously obtained interpolated reference maps through point-wise Kalman filters. This approach enhances accuracy and compensates for stereo camera limitations while preserving data density.

The current validation experiments are designed to evaluate the proposed method in controlled, semi-controlled, and unstructured off-road environments. While these tests reflect typical conditions encountered in off-road scenarios, they do not account for all potential extremes, such as muddy or reflective surfaces, or the effects of heavy rain or snowfall. Under such conditions, the accuracy of both sensors can be affected. For instance, LIDAR signals may be absorbed or scattered on muddy or reflective surfaces, and stereo cameras may face challenges with poor lighting or texture-less environments. Addressing these challenges remains a research question in the future.

Future work should explore a real-time online implementation but could also investigate extending interpolation along LIDAR measurements to increase correction capabilities and reduce smudging effects. It would also be interesting to further evaluate the methodology in more complex environments to explore the impact of reflective or absorbent surfaces and harsh weather conditions. Additionally, investigating a single estimation for the Kalman filter across the entire map could offer further improvements.

## Figures and Tables

**Figure 1 sensors-25-00509-f001:**
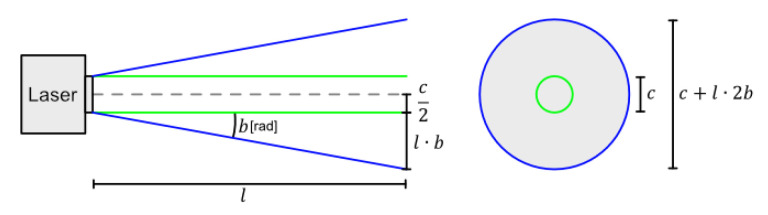
Beam distribution dependant on distance from sensor.

**Figure 2 sensors-25-00509-f002:**
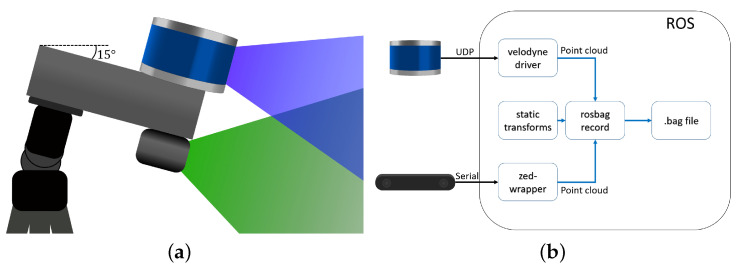
Sensor and software setup. (**a**) Visualizing of how the LIDAR and stereo camera were mounted; (**b**) software setup for recording data.

**Figure 3 sensors-25-00509-f003:**
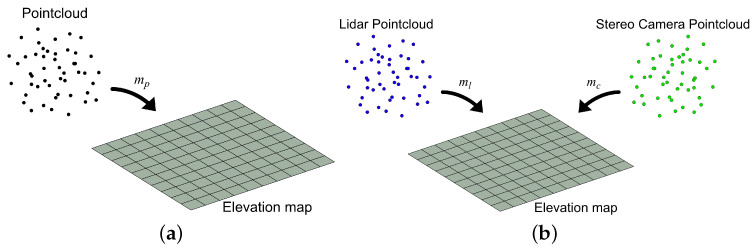
Process of mapping point clouds to elevation map. (**a**) Single point cloud mapping to elevation map; (**b**) multiple point clouds mapping to elevation map.

**Figure 4 sensors-25-00509-f004:**
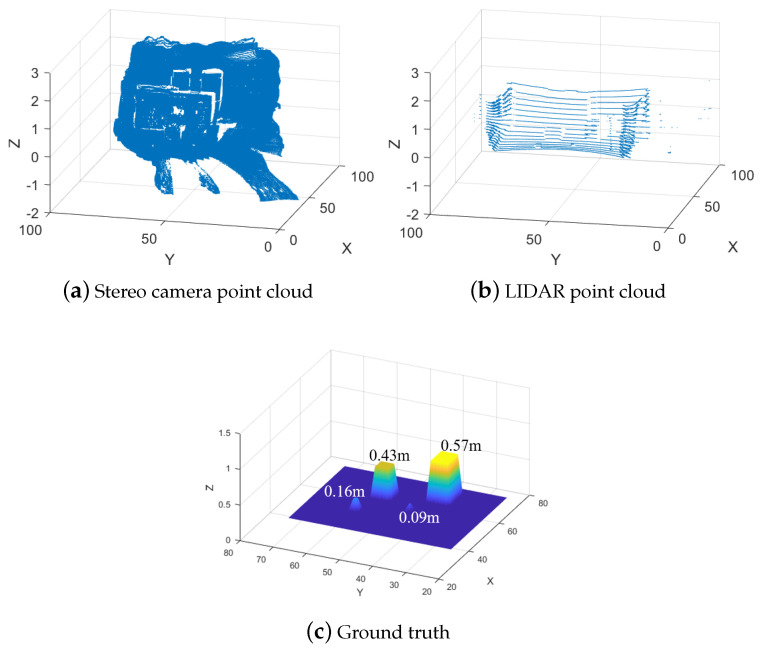
Example of point clouds from stereo camera (**a**), LIDAR (**b**), and the actual ground truth at the center of the point cloud (**c**). Note that (**c**) is zoomed in.

**Figure 5 sensors-25-00509-f005:**
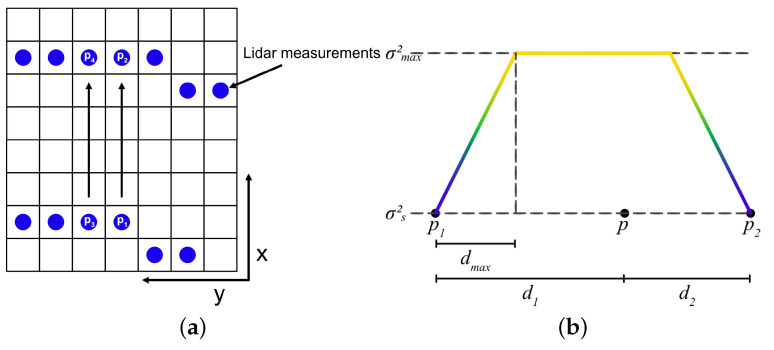
Interpolation methodology. (**a**) Description of the interpolation direction in the gridmap. (**b**) Trapezoid function for variance between two measured points, $p_{1}$ and $p_{2}$.

**Figure 6 sensors-25-00509-f006:**
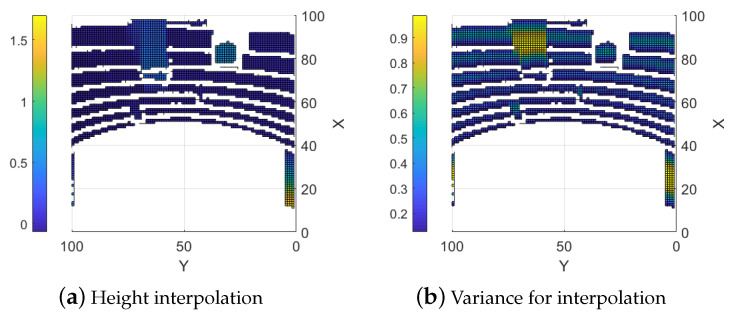
Interpolated map and its corresponding variance. x and y are grid cell indexes.

**Figure 7 sensors-25-00509-f007:**
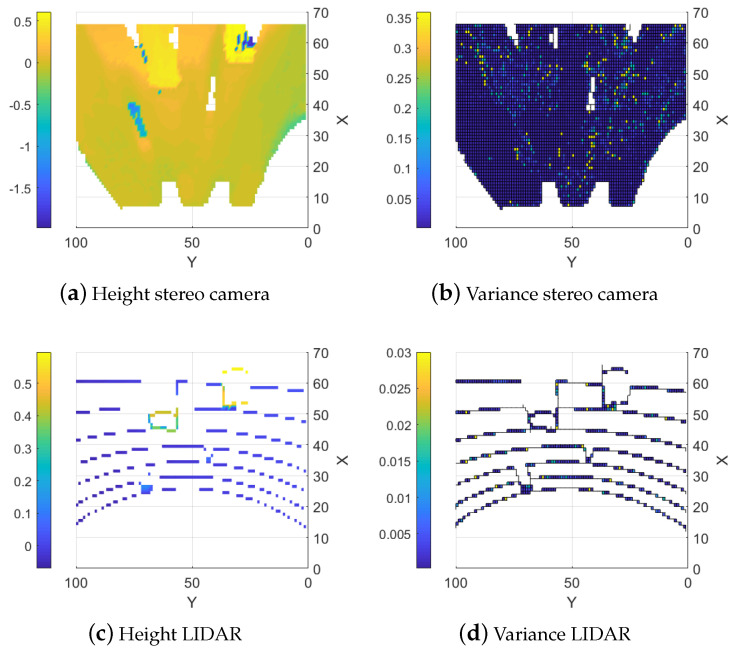
Single sensor elevation map.

**Figure 8 sensors-25-00509-f008:**
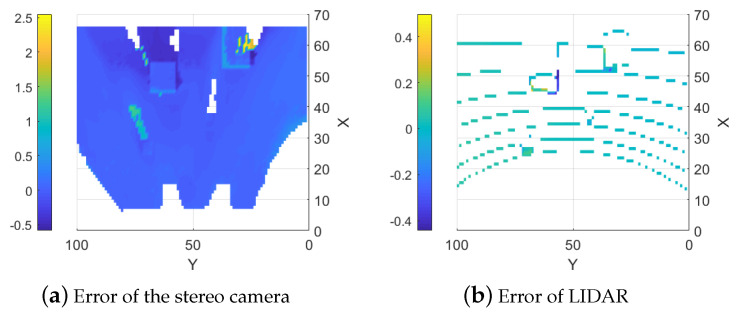
Estimation Errors of the Two Sensors.

**Figure 9 sensors-25-00509-f009:**
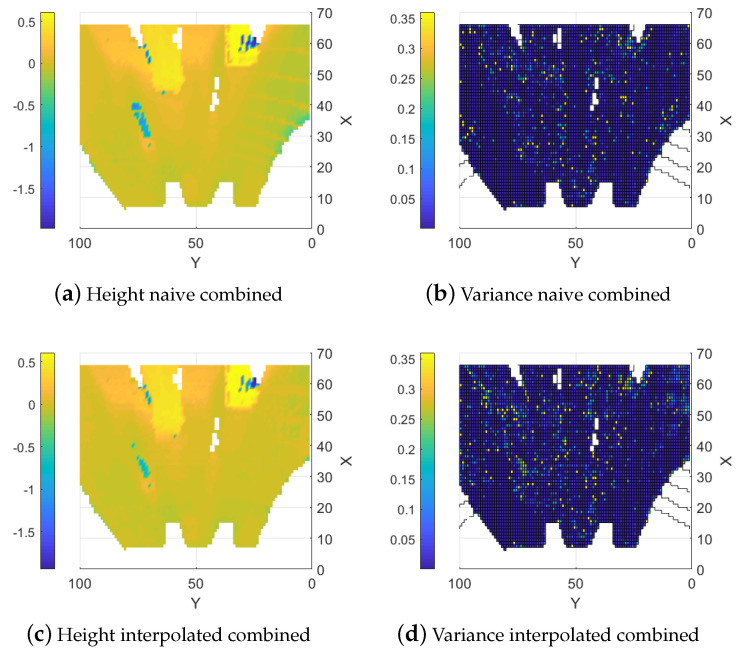
Fused elevation maps.

**Figure 10 sensors-25-00509-f010:**
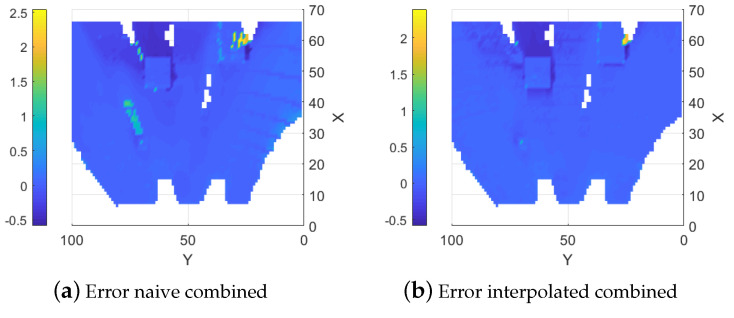
Estimation errors of the two fusion methods.

**Figure 11 sensors-25-00509-f011:**
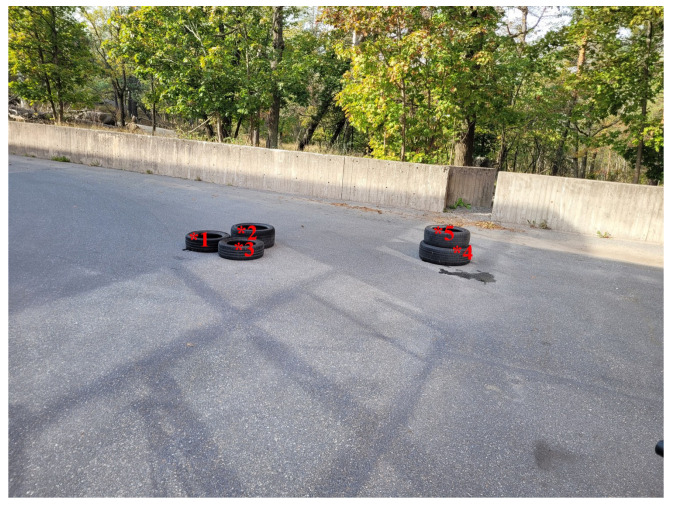
Photograph of the test area, with critical measurement points marked.

**Figure 12 sensors-25-00509-f012:**
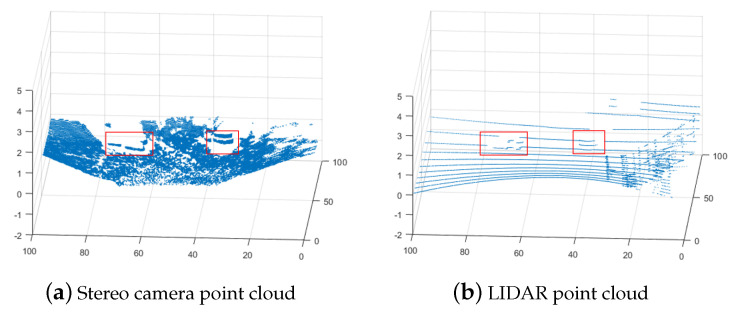
Example of raw point clouds with the objects highlighted.

**Figure 13 sensors-25-00509-f013:**
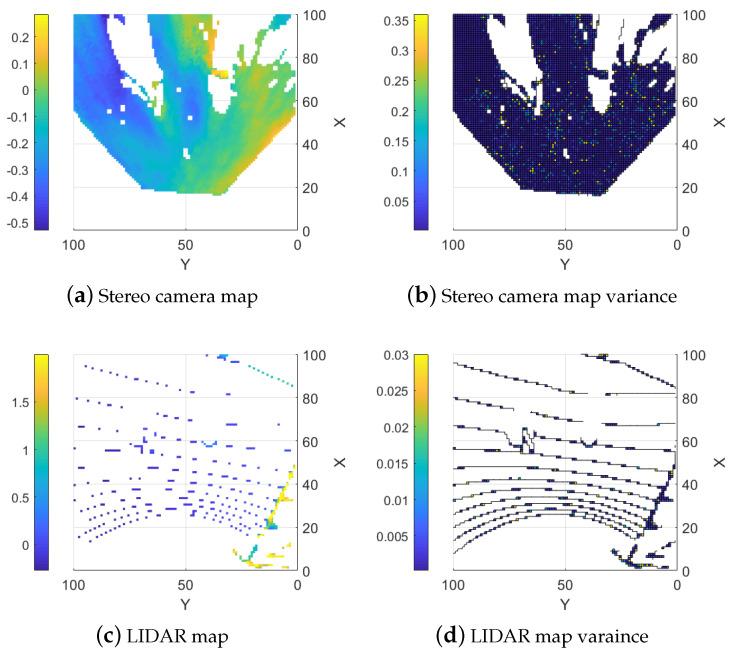
Stereo camera and LIDAR maps with their resulting variance.

**Figure 14 sensors-25-00509-f014:**
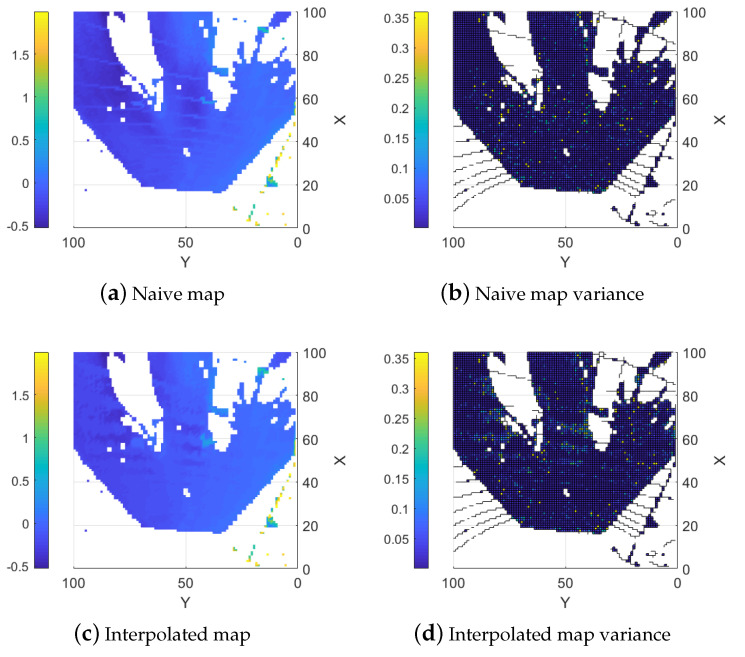
Fused maps with their resulting variance.

**Figure 15 sensors-25-00509-f015:**
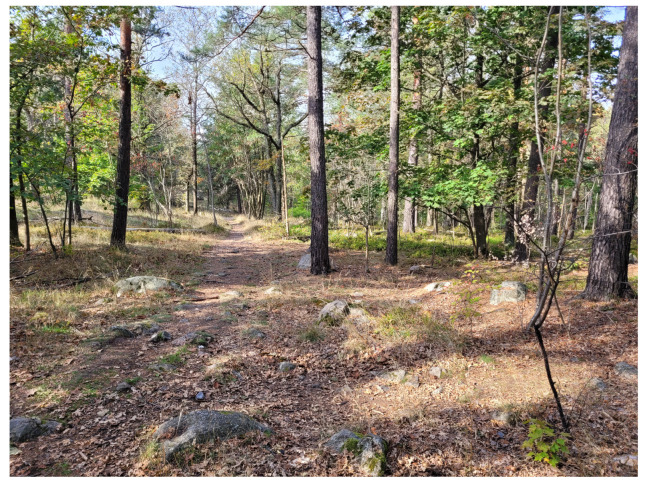
Photograph of the test area.

**Figure 16 sensors-25-00509-f016:**
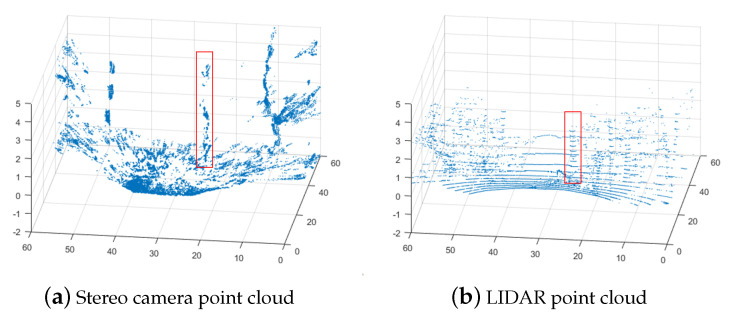
Example of raw point clouds.

**Figure 17 sensors-25-00509-f017:**
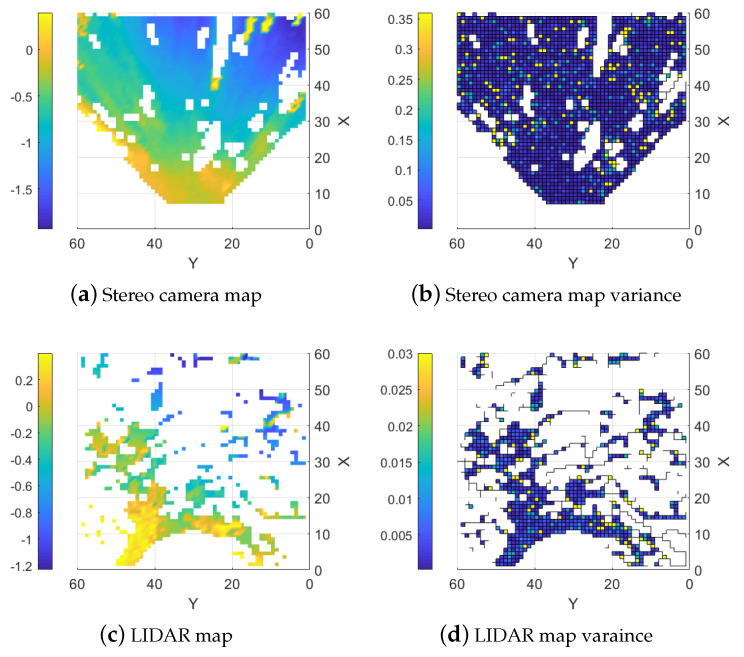
Stereo camera and LIDAR maps with their resulting variance.

**Figure 18 sensors-25-00509-f018:**
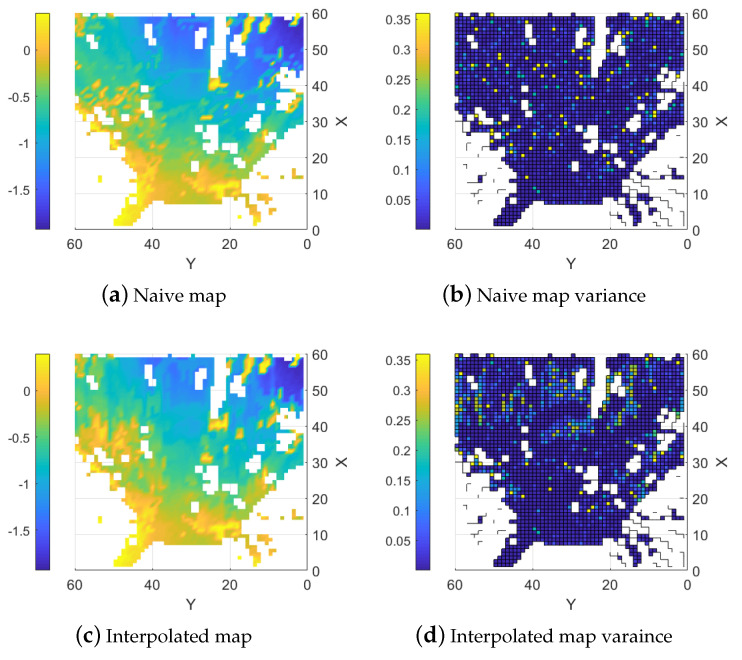
Fused maps with their resulting variance.

**Figure 19 sensors-25-00509-f019:**
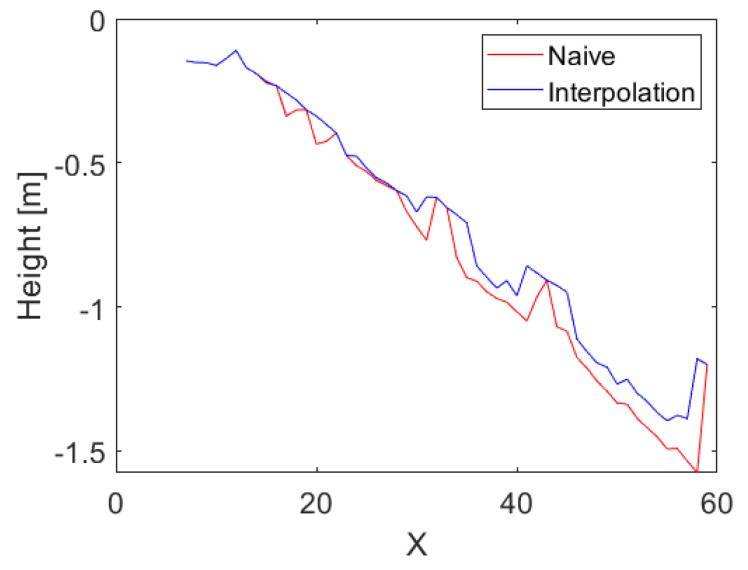
Estimation of both fusion methods along Y = 34.

**Table 1 sensors-25-00509-t001:** Precision and fill percentages of the different methods.

Type	Max Error	Mean Error	RMSE	Fill %
Stereo	2.5496	0.1012	0.2234	62.19%
LIDAR	**0.4984**	**0.0339**	**0.0686**	16.59%
Naive	2.5436	0.0918	0.2100	**63.39**%
Interpolation	2.3913	0.0555	0.1324	**63.39**%

**Table 2 sensors-25-00509-t002:** Heights of measurement points.

Points	1	2	3	4	5
Height [m]	0.185	0.185	0.27	0.24	0.46

**Table 3 sensors-25-00509-t003:** Precision of the different methods.

Type	Stereo	LIDAR	Naive	Interpolation
Point 1 error	0.155	**0.015**	0.025	0.025
Point 2 error	0.095	**0.005**	0.025	**0.005**
Point 3 error	0.1	**0.03**	0.07	**0.03**
Point 4 error	0.14	**0.01**	0.14	0.02
Point 5 error	0.09	0.01	0.03	**0.0**
Max error	0.16	**0.03**	0.14	**0.03**
Mean error	0.12	**0.01**	0.06	0.02
RMSE	0.13	**0.02**	0.07	**0.02**

## Data Availability

The original contributions presented in this study are included in the article material. Further inquiries can be directed to the corresponding authors.

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
