# Peer review of "Enhancing Off-Road Topography Estimation by Fusing LIDAR and Stereo Camera Data with Interpolated Ground Plane"

_sensors, 2025, doi:10.3390/s25020509_

Round 1
Reviewer 1 Report
Comments and Suggestions for Authors
Here are some suggested strengths and weaknesses to revise the article "Enhancing Off-Road Topography Estimation by Fusing LIDAR and Stereo Camera Data with Interpolated Ground Plane," with recommendations for major revision to make it publishable. Addressing these aspects could enhance this article's clarity, rigor, and Sensors or similar journals standard nobility.
Strengths
- The paper introduces a novel approach to constructing an interpolated ground plane using LIDAR data to improve stereo camera accuracy, contributing to sensor fusion for autonomous off-road navigation.
- The research comprehensively evaluates the proposed method across controlled, semi-controlled, and unstructured environments, enhancing the robustness and applicability of the findings.
- Sections such as the baseline and interpolation-based approaches are well-detailed, and the use of diagrams (Figures 5, 6) effectively clarifies the methodology.
- The inclusion of precision metrics, such as RMSE and mean error comparisons, strengthens the quantitative validation of the proposed method, which is crucial for establishing credibility.
General comments
1. Abstract:
o The abstract should be stand-alone, and acronyms should be written in full name and abbreviation in brackets (e.g., LIDAR)
o The abstract should include a brief intro, method, result (supported with analysis) and conclusions. The methodology you applied was not presented briefly. For example, authors mentioned” improves accuracy in homogeneous areas” line 12 but no how much is it. Can you quantify it? Similarly, “performs better at greater distances” in line 13, but not quantified here. Can you quantify it?
2. Introduction part
o I suggest adding a section explaining the choice of Kalman filter parameters, its advantages over alternatives like Extended Kalman Filters or Particle Filters, and its implications on computational complexity and accuracy.
o As minor edits, write full name or basic definition of acronyms in their first appearance (e.g., 2D, 2.5 D, 3D, and TOF). Check and fix such kind of error through your manuscript.
3. Methodology part
o While the RMSE, mean error, and max error provide insight into the method's performance, statistical tests are lacking in validating the significance of improvements. Thus, including statistical tests, such as paired t-tests, could strengthen claims about the fusion method's effectiveness compared to individual sensor data, especially in diverse environmental conditions.
o The study lacks an analysis of real-time processing feasibility, mainly because autonomous navigation requires rapid data processing. Therefore, it is recommended that the computational load and real-time processing potential be discussed. If feasible, a section on optimizing for real-time implementation could be added.
o The paper focuses heavily on off-road navigation without exploring other applications, limiting the research's broader relevance. Therefore, briefly mention or suggest applications beyond off-road navigation, such as in agriculture or disaster response, to broaden the appeal.
4. Result part
o Briefly, elaborate what “x” and “y” represent in Figures 7−10, 13−14, and 17−18. What do the different colors of the legend represent in the figures?
o Check uncited figures in text. For example, Figures 3(a), 6(b), 7(b−d), 8(a−b), 9(b−d), and 10a. Thus, your analysis need further discussion and scientific explanation for each analysis that should be supported with further references in the discussion part or result part.
5. Conclusions and Limitation of the study
o While the article highlights the benefits and drawbacks of each sensor, it does not sufficiently address how their specific limitations (e.g., stereo camera sensitivity to lighting) were managed in experiments in terms of sensor-specific challenges. Thus, add more details on experimental controls or adjustments made to mitigate known sensor-specific challenges, potentially improving reproducibility for readers.
Comments on the Quality of English Language
The English could be improved to more clearly express the research.
Author Response
Please see our response in the attached file.

Reviewer 2 Report
Comments and Suggestions for Authors
This paper focuses on topography map estimation in off-road scenarios. The authors first construct a reference ground plane through interpolation of LIDAR data, which is then fused with stereo point cloud via Kalman Filter. Practical experiments are performed to validate the proposed methods. Some concern should be resolved:
The Introduction section contains many plain expressions without strong logical connections. Please make effort to polish and improve it.
A common conclusion is that stereo cameras has denser scanning than LiDAR, while the latter has higher accuracy. But this is not absolute. It depends on sensor settings and stereo matching algorithms. Please give more references or evidence to support this conclusion instead of wearisome statements. The two papers are citable, which give valuable insights in terms of sensor, data, and experiment results: doi.org/10.1038/s41597-024-03261-9, doi.org/10.1109/TITS.2024.3431671.
In the Related Works section, the authors should give more brief and summative descriptions to existing research. Demonstrate the general schemes and bottleneck problems.
I have not seen any details about stereo matching for disparity regression and recovering point clouds. The authors should give more details and evaluations about point cloud’s quality.
The proposed scheme for fusion may not be robust to complex scenarios. The validation experiments are not sufficient. Please consider and discuss scenarios like rainy roads, texture-less roads.
It seems that experiments are conducted in static state. How is the performance when sensors are moving? This is crucial for autonomous vehicles and robots.
Comments on the Quality of English LanguageMake more efforts to revise and polish it.
Author Response
Please see our responses in the attached file.

Reviewer 3 Report
Comments and Suggestions for Authors
The authors can expand the introduction section by introducing a more substantial pros and cons discussion of LIDAR and Stereo Imaging, possibly in a table. A plus would be the introduction of physical systems for commercial applications.
The authors can discuss the implications of using LIDAR vs Camera based sensing in a system engineering scenario, dealing with increased/decreased ocmplexity, power implications, data flow, reliability.
How are the algorithms evaluated? Did the authors use some particular software, or did they run the algorithms in embedded systems? the authors should explain better how the sensing data is processed to obtain the enchanced results.
Is the processing executable in real time?
How does this method compare to other in therms of quantitative results?
In fig.4 please uniform the axis notations.
Author Response

(The authors gave the same response as above.)

Round 2
Reviewer 1 Report
Comments and Suggestions for Authors
All my comments and suggestions are addressed by authors; I have no further comments.
Comments on the Quality of English LanguageThe English could be improved to more clearly express the research.